# The Effect of Baking Heat Treatment on the Fatigue Strength and Life of Shot Peened 4340M Landing Gear Steel

**DOI:** 10.3390/ma13245711

**Published:** 2020-12-15

**Authors:** Seok-Hwan Ahn, Jongman Heo, Jungsik Kim, Hyeongseob Hwang, In-Sik Cho

**Affiliations:** 1Department of Aero Mechanical Engineering, Jungwon University, Chungbuk 28024, Korea; 2Aerospace Department, EM KOREA Co., Ltd., Changwon 51538, Korea; jmheo@yesemk.com (J.H.); jskim@yesemk.com (J.K.); hshwang@yesemk.com (H.H.); 3R&D Team, Mbrosia Co., Ltd., Asan 31460, Korea

**Keywords:** 4340M, shot peening, baking heat treatment, ultrasonic fatigue test, inclusion

## Abstract

In this study, the effect of baking heat treatment on fatigue strength and fatigue life was evaluated by performing baking heat treatment after shot peening treatment on 4340M steel for landing gear. An ultrasonic fatigue test was performed to obtain the S–N curve, and the fatigue strength and fatigue life were compared. The micro hardness of shot peening showed a maximum at a hardened depth of about 50 μm and was almost uniform when it arrived at the hardened depth of about 400 μm. The overall average tensile strength after the baking heat treatment was lowered by about 80–111 MPa, but the yield strength was improved by about 206–262 MPa. The five cases of specimens showed similar fatigue strength and fatigue life in high cycle fatigue (HCF) regime. However, the fatigue limit of the baking heat treated specimens showed an increasing tendency rather than that of shot peening specimens when the fatigue life was extended to the very high cycle fatigue (VHCF) regime. The effect of baking heat treatment was identified from improved fatigue limit when baking heat was used to treat the specimen treated by shot peening containing inclusions. The optimum temperature range for the better baking heat treatment effect could be constrained not to exceed maximum 246 °C.

## 1. Introduction

Recently, demands on the long life reliability and endurance evaluation of aircraft component materials are rapidly increasing. Advanced research institutes and companies of component materials have developed various test methods about acceleration, and the evaluation and analysis are under big progress. Elastic modulus, Poisson’s ratio and fatigue characteristics are among the most important and basic properties, but they have disadvantages of difficulty and long period in precise testing. Thus, the simple and accurate measurement technique of dynamic elastic modulus and Poisson’s ratio using a characteristic frequency of acoustic resonance evaluation technique of fatigue test using ultrasonic elastic resonance in a short period of time were world-widely applied in many component materials [1,2,3]. Light- and high-strength materials are required as aircraft materials, and ferrous materials are used in main components. Especially AISI 4340 [4], 4340M [5] and 300M [6] steels are used in aircraft landing gear components. These materials are super high-strength materials with tensile strength above 2000 MPa, and high strength and toughness are very important in aircraft landing gear materials [6]. Aircraft landing gear needs to resist impact energy when landing on and high specific strength is required in order to obtain anti-fatigue and weight lightening during when taxiing and when taking off and landing. Therefore, aircraft landing gear materials are treated by shot peening (SP) for fatigue resistance [7]. SP is a widely used technique where the surface is hardened by compressive residual stress imposed by shot impact of cold working process and fatigue strength and fatigue life are improved [8]. Aircraft landing gear materials are baking heat treated as a post heat treatment process to reduce hydrogen embrittlement after SP. The baking heat treatment can prevent hydrogen embrittlement [9,10,11], but it might reduce the fatigue strength improved by SP. According to the standard overhaul practices manual, the baking heat treatment to minimize the decrease in residual stress by SP is suggested to be conducted not exceeding 246 °C [12,13]. However, the precise control of the baking heat treatment temperature in aircraft landing gear maintenance, the repair and overhaul (MRO) process is difficult. Therefore, in this study, in the baking heat treatment performed as a post heat treatment process to relieve hydrogen embrittlement after SP treatment, it was experimentally investigated whether the fatigue strength and lifespan would not be affected by the heat treatment in a certain temperature range. The purpose of this study was to evaluate the effect of baking heat treatment on the fatigue strength in a long cycle life by a ultrasonic fatigue test using 4340M steel, which is an SP treated aircraft landing gear material. In addition, it was also tried to determine how the effect of inclusions related to crack initiation had a certain correlation with SP and baking heat treatment.

## 2. Experiments

### 2.1. Specimen Preparation

The 4340M steel used in this study was Cr-Ni-Mo-V steel with ultrahigh strength and high resistance to fatigue with chemical compositions as shown in Table 1. The composition analysis of 4340M steel was analyzed according to KS D 1652 [14]. The 4340M steel is actually a landing gear material used for the B737 and A320 civil aircraft. Specimens were prepared in five conditions to evaluate the effect of baking heat. For the relative comparison of each condition, the five cases of specimens of quenched tempering (QT) standard treated specimen (STD), shot peened (SP) specimen, and baking heat treated specimens such as SP-200 (200 °C), SP-B246 (246 °C) and SP-B260 (260 °C) were prepared. Figure 1a,b show the heat treatment process. The furnace for the quenching heat treatment was used C-01 (2000ENG Co., Incheon, Korea) with 100 kW of an electric capacity, 760 × 1220 × 650 mm of a heating area and 1000 kg of a capacity as an air circulation method. The furnace for the tempering heat treatment was used T-06 (2000ENG Co.) with 70 kW of an electric capacity, 760 × 1220 × 650 mm of a heating area and 1000 kg of a capacity as an air circulation method. Moreover, the furnace for the baking heat treatment was used, TM-51 (Joongang Co., Cheongju, Korea), with 50 kW of an electric capacity, 1000 × 1000 × 800 mm of a heating area and 1000 kg of a capacity as an air circulation method. Figure 1c–e shows the appearance of the used heat treatment furnace. The base material of 4340M steel was quenched for 2 h at 850 °C (Figure 1a) and then followed by tempering for 3 h at 160 °C (Figure 1b). The baking heat treatment was, respectively, conducted for 4 h at 200, 246 and 260 °C after SP on heat treated specimens as shown in Figure 1a.

The 4340M steel is a hypoeutectoid alloy of 0.43% C with a combination of proeutectoid ferrite and pearlite microstructures. This proeutectoid microstructure has the advantage of high vibration absorption and is proper to aircraft landing gear material such as 4340M steel.

The SP in this study was conducted following the SP specifications of standard overhaul practices, manually applied to aircraft landing gear materials. SP conditions applied to aircraft landing gear material are shown in Table 2 [12,13].

### 2.2. Tensile, Hardness, and Ultrasonic Fatigue Test

The specimens for micro hardness were manufactured by cutting from five conditions of fatigue specimen. It was measured by applying a load of 200 g and a load time of 15 s using micro-Vickers hardness tester, MICROMET (Buehler Co., Lake Bluff, IL, USA). The tensile test was conducted with 4340M steel prepared from five conditions. Bar type specimen was manufactured along ASTM E8 [15] with a gage length of 50 mm and a diameter of 12.5 mm as shown in Figure 2. Universal testing machine (UH-F100A, Shimadzu Co., Kyoto, Japan) of a maximum load of 1000 kN was used for the test with a cross-head speed of 3 mm/min in air at room temperature. Tensile test results were obtained using three specimens according to each condition, respectively. Based on the tensile test results, the stress range for the fatigue test was determined.

The basic theory of ultrasonic fatigue test used in this study is the resonance test method using PZT and 20 kHz elastic vibration wave producing maximum strain at the middle of gage length by maximum displacement at the end of specimen resulting maximum strain amplitude as shown in Figure 3 [1,2,3,16]. The ultrasonic fatigue test machine used in the experiment was Y-UFO (Mbrosia Co., Asan, Korea) of 20 kHz capacity [2]. Table 3 shows the results of dynamic Young’s modulus (E), shear modulus (G), and Poisson’s ratio (μ) of 4340M steel based on ASTM E1876 specification [17]. I-BAT (Mbrosia Co., Asan, Korea) was used for the measurement [2]. Two specimens were used to obtain the dynamic Young’s modulus, shear modulus, and Poisson’s ratio. Dynamic Young’s modulus and Poisson’s ratio were measured as 215.12 GPa and 0.266, respectively, and were used in specimen design and the determination of stress for the ultrasonic fatigue test.

Specimens were designed to have the same gage length and total dimension for five conditions considering dynamic elastic modulus of 4340M steels, and the female tap was manufactured for the specimen to be jointed to the resonance horn as shown in Figure 4. The fatigue specimen was precisely polished to a surface roughness of R_a_ 0.02 μm level to minimize the effect of surface roughness, which might be an important parameter. In addition, after the fatigue test, the fatigue fracture surfaces were observed by scanning electron microscopy (SEM, Jeol-6510, Jeol Co., Tokyo, Japan).

## 3. Results and Discussion

### 3.1. Hardness and Tensile Test Results

Figure 5 shows the result of micro hardness measurement along the depth from the surface. The STD specimen showed a micro hardness of HV 590 with some kind of scatter plot. The SP specimen showed an average micro hardness of HV 650 at depth of about 50 μm away from surface and almost was uniform when HV 600 arrived at the hardened depth of about 400 μm. Three cases of baking heat treated specimens an showed almost similar tendency with the SP specimen. The SP-B260 specimen showed the highest hardness value near the surface among them. This is thought to come from grain refinement due to the compressive residual stress effect caused by the surface plastic deformation by SP. Below the hardened depth of about 200 μm, the SP specimen did not show a big difference in the hardness distribution with the baking heat treated specimens, and this was because the hardness was kept by the baking heat treatment instead of decreasing [18]. Above the hardened depth of about 200 μm, the hardness of SP-B246 specimen generally showed the largest one. This is thought to come from the bake hardening effect by the baking heat treatment which depresses the hardness decrease. Therefore, the optimum temperature range for the baking heat treatment after SP was 246 °C where the hardness decrease is depressed. On the other hand, both SP-B200 and SP-B260 specimens showed a larger variation range in the hardness distribution. Therefore, this means that the optimum temperature for baking heat treatment after SP is 246 °C when judged from hardness distribution.

Table 4 shows the mechanical properties of 4340M steel obtained from the tensile test. Figure 6 shows the stress–strain curve. The tensile strength of the SP specimen is higher than that of the heat treated base material (STD). While the tensile strength obtained by the baking heat treatment after SP reduced compared to SP only, on the other hand, the yield strength of the specimens by the baking heat treatment increased compared to that of STD and SP specimens, and showed the largest value at a baking heat treatment temperature of 246 °C. This result was identified as an effect of the increasing yield strength and elongation by reducing the hydrogen embrittlement by the baking heat treatment, which will directly affect the fatigue strength in the elastic region.

### 3.2. S-N Curve and Fracture Surface Observation

Figure 7 shows the S–N curves of the five cases of specimens obtained from the ultrasonic fatigue test. From the results of the ultrasonic fatigue test, the fatigue strength and fatigue life of the five cases of specimens did not show a significant difference below the high cycle fatigue (HCF) regime. However, the fatigue limit of the baking heat treated specimens increased in the very high cycle fatigue (VHCF) regime with low fatigue stress. This can be identified from the tensile test results which showed similar yield strength in both STD and SP specimens, but showed higher yield strength values in the three cases of baking heat treated specimens when the yield strength was considered as a direct parameter among the mechanical properties.

Fatigue limit of the baking heat treated specimens showed an increased tendency compared to that of the STD and SP specimens when the fatigue life was extended to the VHCF regime. Thus, the fatigue fracture surfaces of all specimens were observed to investigate the reason why the SP specimen showed a lower fatigue limit than baking the heat treated specimens.

Fatigue fracture surfaces are shown in Figure 8a–e. Figure 8 shows the overall phenomenon of fatigue failure fractured by inclusion, and from the surface. As a result of analyzing the fatigue fracture surfaces, the main reason for the effect on fatigue strength and fatigue life is not that the origin of crack initiation by fatigue begins the near of surface or the interface with the hardened layer formed by the SP, but that it begins in the form of an internal crack due to the influence of inclusions contained in the virgin material. In other words, the origin of crack initiation appeared, which mainly occurred as the form of the internal crack initiation from the interior inclusions. When the inclusion was not present, the crack initiation started near the interface through SP hardening. However, when the inclusion was contained, the crack initiation started from the inclusion almost with the form of an internal crack. A fine granular area (FGA) was found and formed in the vicinity around the inclusion [18,19,20]. A typical fish eye could be observed on the fracture surface, which was initiated from the inclusion. Specially, as the fatigue cycle is increased to a long life, which is longer than 10^6^ cycles, the origin of crack initiation showed a tendency to occur from the interior part of the material instead of near the interface by the peening hardening effect.

First of all, therefore, it is preferred to use clean material without inclusions in order to maximize the effect of severe surface plastic deformation by SP to increase fatigue life. That is, it is best to use materials without any inclusions in order to increase the fatigue limit by the compressive residual stress effect formed by SP. However, if SP is performed when the material contains inclusions, the stress concentration is applied to the inclusions by SP resulting as a cause of the lowering fatigue limit [18,19,20]. However, as the stress applied to the inclusions by SP is relaxed through the baking heat treatment performed after SP, the stress concentration causing fatigue is relieved and the fatigue limit is not lowered. Therefore, it is considered that the effect of the baking heat treatment is exhibited because the fatigue limit obtained by the baking heat treatment is higher than one in the case of SP on a material with inclusions.

The baking heat treatment produced a higher fatigue limit in 246 °C (SP-B246) specimen than the 200 °C (SP-B200) and 260 °C (SP-B260) specimens. This result shows that it is necessary to keep the baking heat treatment temperature not to exceed 246 °C for the best effect. In addition, the baking heat treatment temperature range of 200 °C is not recommended. Moreover, in the temperature range of 260 °C, where the temperature is somewhat higher, it was found that the fatigue limit was slightly higher than one of 200 °C, but almost similar values were formed. Therefore, the optimum temperature range for a better baking heat treatment effect could be constrained not to exceed maximum 246 °C, and it is reasonable to limit the baking heat treatment temperature after the SP of 4340M steel for landing gear required in aircraft maintenance, repair and overhaul (MRO) process.

In the MRO process, it is difficult to match the baking heat treatment temperature in the field, so the aim of this study was to test the ambient temperature range of 246 °C. In the next study, we think it is necessary to conduct additional fatigue tests for a more detailed temperature range. In future plans, by using the results in this study, authors also intend to evaluate fatigue strength and life using specimens which conduct chrome plating after shot peening treatment.

Figure 9 shows the composition analysis on inclusions in STD, SP and SP-B246 specimens. The overall size of inclusion was found to be about 5–10 μm, and a mapping result around the inclusion showed increasing Ni and Al. And Ni, S and Ca were observed as main components by the precise analysis of the inclusion through enlarged Figure 8b.

## 4. Conclusions

The 4340M landing gear steel was treated by SP to improve fatigue strength followed by baking heat treatment as a post heat treatment to reduce hydrogen embrittlement. The results of the study on the effect of baking heat treatment on the improvement of the fatigue strength and fatigue life of landing gear steel in very high cycle regime and the optimum baking heat treatment temperature can be summarized as follows:Composition and microstructure analyses showed that 4340M landing gear steel is a hypoeutectoid alloy with 0.43% C containing Ni which maximizes the effect of heat treatment temperature control and has ultrahigh strength and good toughness by the vibration absorption through a mixed microstructure of proeutectoid ferrite and pearlite.The surface micro hardness of a heat-treated specimen (STD) was HV 590. SP showed a maximum HV 650 at a hardened depth of about 50 μm and almost uniformed as HV 600 when it arrived at the hardened depth of about 400 μm. Additionally, three cases of baking heat treated specimens showed an almost similar tendency with the SP specimen. The SP-B260 specimen among them showed the highest hardness value near the surface.The 4340M steel showed the dynamic Young’s modulus of 215.12 GPa and Poisson’s ratio of 0.266.From the results of the tensile test, the STD specimen showed an average tensile strength of 2098 MPa and an average yield strength of 1360 MPa, and SP specimen showed an average tensile strength of 2129 MPa and an average yield strength of 1416 MPa, respectively. The overall average tensile strength after the baking heat treatment was 2018 MPa, and it was lowered by about 80–111 MPa. However, the yield strength after the baking heat treatment was 1504–1622 MPa, and it was improved by about 206–262 MPa.From the ultrasonic fatigue test result, the five cases of specimens showed similar fatigue strength and fatigue life in the HCF regime. However, the fatigue limit of the baking heat treated specimens showed an increased tendency compared that of the STD and SP specimens when the fatigue life was extended to the VHCF regime.The effect of baking heat treatment was identified from an improved fatigue limit when we used the baking heat treatment for the specimen treated by SP containing inclusions. It is necessary to keep the optimum baking heat treatment temperature under exceed 246 °C, and the temperature range of 200 °C or 260 °C is not recommended. Therefore, the optimum temperature range for the better baking of the heat treatment effect could be constrained not to exceed a maximum of 246 °C, and it is reasonable to limit the baking heat treatment temperature after the SP of 4340M steel for the landing gear required in an aircraft MRO process.

## Figures and Tables

**Figure 1 materials-13-05711-f001:**
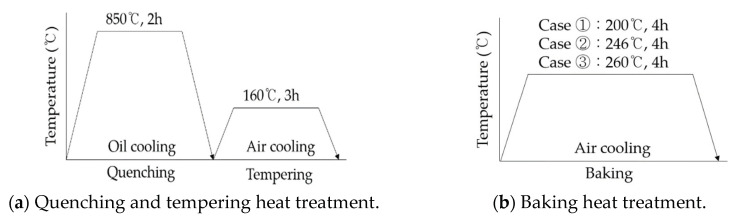
Heat treatment processes and the appearances of used furnace.

**Figure 2 materials-13-05711-f002:**
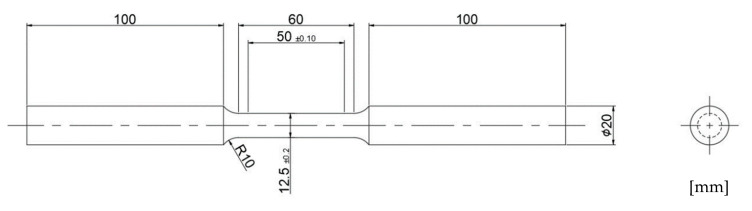
Standard 12.5 mm round tension test specimen of ASTM E8.

**Figure 3 materials-13-05711-f003:**
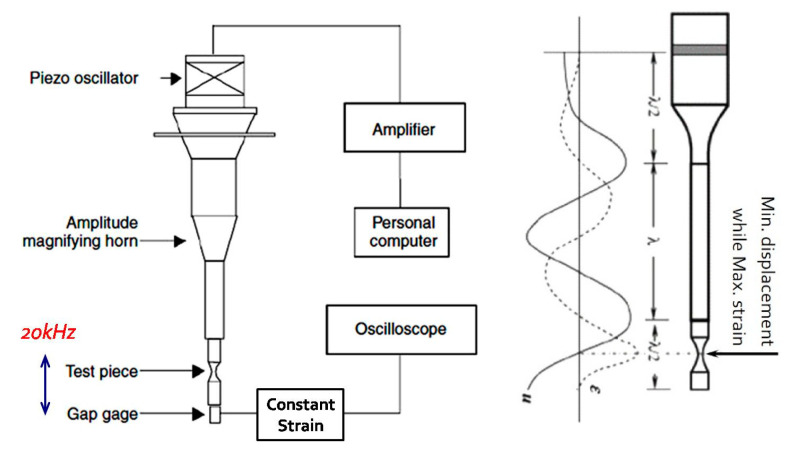
Schematic diagram of ultrasonic fatigue test [2,15].

**Figure 4 materials-13-05711-f004:**
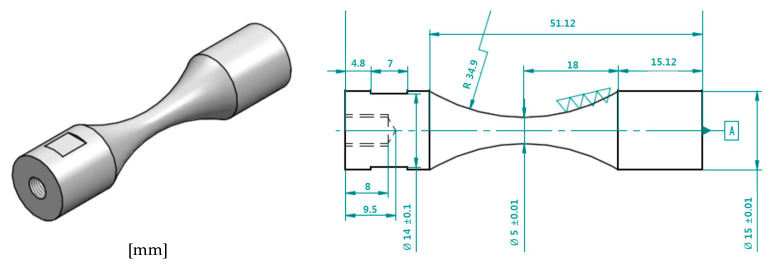
The shape of the fatigue specimen of 4340M steel manufactured considering the dynamic elastic modulus and density.

**Figure 5 materials-13-05711-f005:**
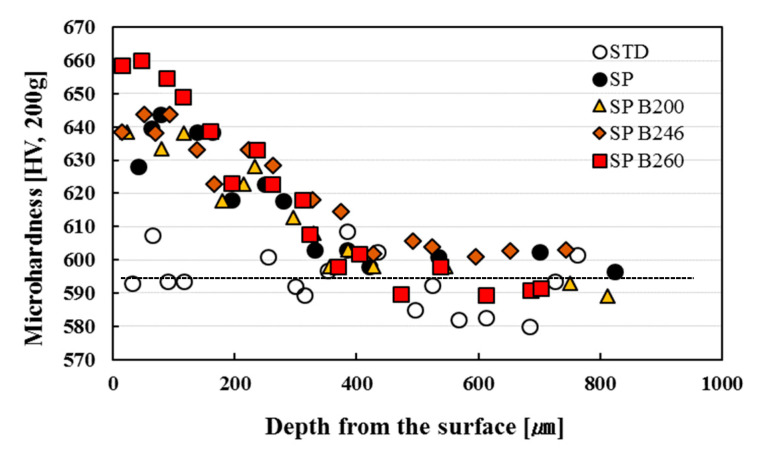
The distribution of micro hardness according to the depth from the surface.

**Figure 6 materials-13-05711-f006:**
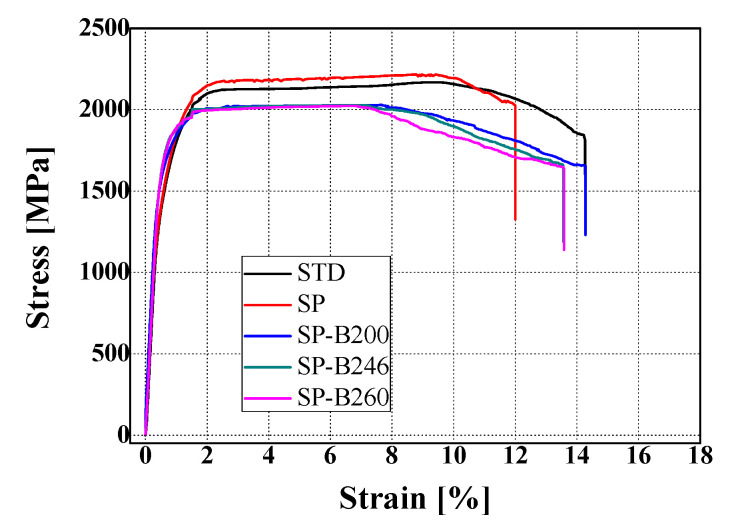
Stress–strain curves of the 4340M steel obtained from tensile test.

**Figure 7 materials-13-05711-f007:**
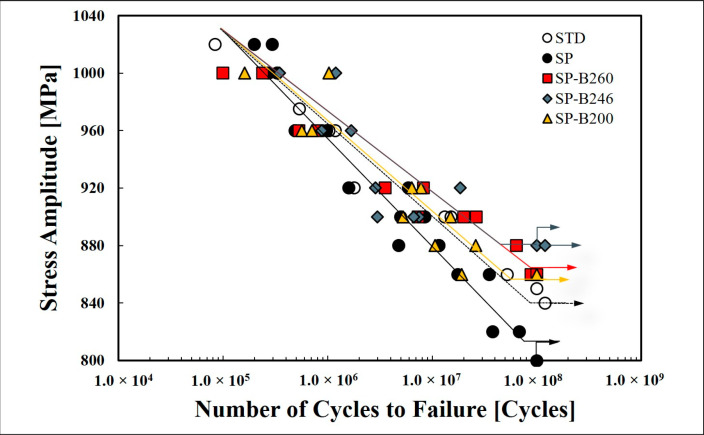
S–N curve obtained from the fatigue test (20 kHz, R = −1).

**Figure 8 materials-13-05711-f008:**
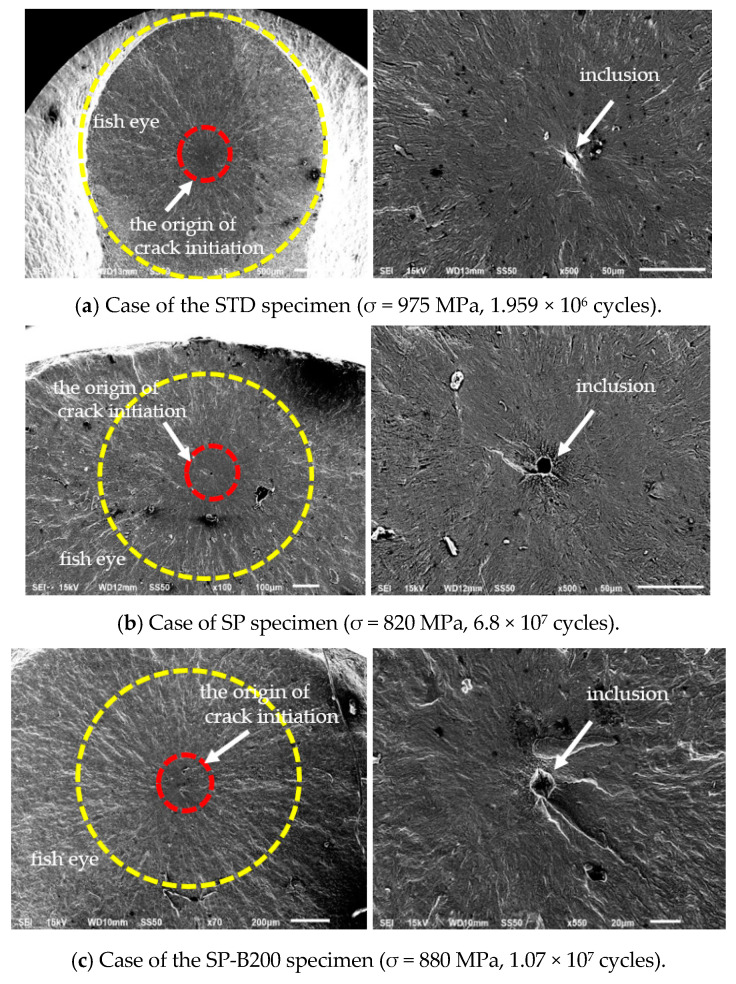
SEM photographs of the fatigue fracture surface obtained from the ultrasonic fatigue test.

**Figure 9 materials-13-05711-f009:**
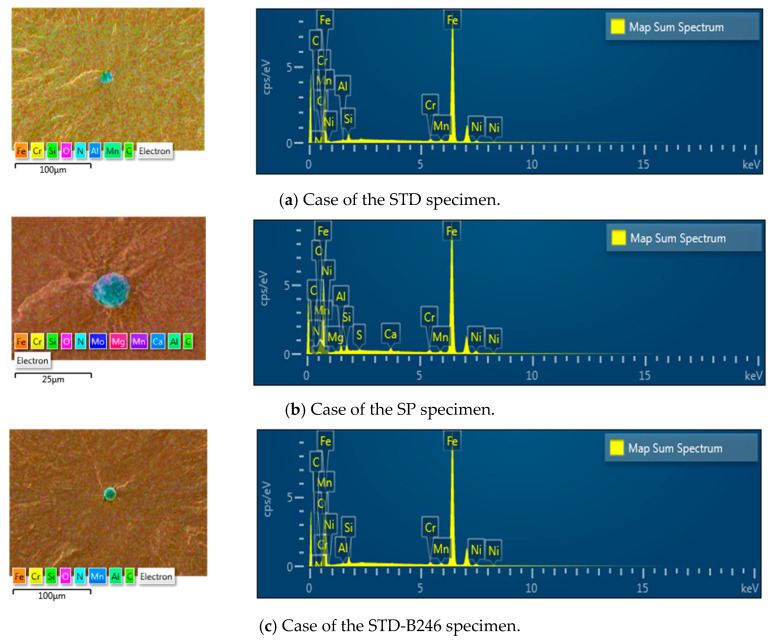
EDS spectra of the inclusion which occurred in the used specimens.

**Table 1 materials-13-05711-t001:** Chemical composition of the 4340M steel (in wt.%). [14].

C	Mn	Si	Cr	Ni	Mo	V	N	Nb
0.43	0.83	1.62	0.81	1.82	0.39	0.07	0.002	0.01
**P**	**S**	**Cu**	**B**	**Ti**	**Al**	**W**	**Co**	**Ca**
0.006	0.001	0.12	0.0002	0.006	0.07	<0.05	<0.005	<0.001

**Table 2 materials-13-05711-t002:** Shot peening (SP) conditions.

Cast Steel Shot	Shot Diameter (mm)	Nozzle Diameter (mm)	Shot Flow (kg/min)	Angle of Impingement (degree)
ASH 230 with HRC 55–60	0.7	7.9	3	45–90
**Air Pressure** **(bar)**	**Peening Time** **(min/place)**	**Working Distance (mm)**	**Arc Height (mmA)**	**Coverage** **(%)**
3	2	50 ± 5	0.36–0.38	200

**Table 3 materials-13-05711-t003:** Test results of the dynamic Young’ s modulus, shear modulus, and Poisson’s ratio.

Density (kg/m^3^)	Dynamic Young’s Modulus E (GPa)	Dynamic Shear Modulus G (GPa)	Poisson’s Ratio µ
7804.14	215.12 ± 0.2	84.96 ± 0.2	0.266

**Table 4 materials-13-05711-t004:** Mechanical properties obtained from the tensile test of 4340M steel.

Specimen	Ultimate Tensile Strength (MPa)	Yield Strength (MPa)	Elongation (%)
STD	2098 ± 70	1360 ± 9	14.3 ± 1.0
SP	2129 ± 85	1416 ± 76	12.4 ± 0.4
SP-B200	2036 ± 7	1504 ± 12	13.5 ± 0.8
SP-B246	2011 ± 16	1619 ± 12	13.6 ± 0.1
SP-B260	2008 ± 15	1622 ± 4	14.0 ± 0.5

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
