# Peer review of "The Effect of Baking Heat Treatment on the Fatigue Strength and Life of Shot Peened 4340M Landing Gear Steel"

_materials, 2020, doi:10.3390/ma13245711_

Round 1

Reviewer 1 Report

In the paper, the effect of baking heat treatment on fatigue properties of the steel used for landing gear is evaluated. The research is aimed at determining the influence of heat treatment temperature and shot peening on mechanical properties of steel under consideration.

In my opinion, the paper deals with an interesting and actual topic, however, some critical points need to be addressed and are listed below:

  1.  The abstract of the paper should be rebuilt. The major part of the abstract takes the form of a detailed summary of the results. I think you should give background to the paper, a brief description of the methods, the principle results, then conclusions without specific values for all findings.
  2. Picture in figure 2 is incorrect in case of the round specimen. Vertical lines at the fillets near higher diameter are missing - compare to figure 4.
  3. I would like to ask how many specimens of each condition have been tested to obtained results presented in table 3 and 4. This information should be specified in the paper text.
  4. The mentioned ASTM E1876 and ASTM E8 standards should be listed in the reference list.
  5. Page 7, line 5: I think there is typo error - "...of over 106 cycles,..." I assume it should be "10cycles".

Author Response

See the attachement

Reviewer 2 Report

The paper deals with a problem of the determining the temperature range during the baking heat treatment after SP, which does not affect fatigue strength and fatigue of the aircraft landing gear. In the introduction is written: “According to the standard overhaul practices manual, the baking heat treatment to minimize the decrease of residual stress by SP is suggested to be conducted not exceeding 246℃ [24,25].” The quoted literature included Boeing and SAE International manuals. It is well known that in the aerospace industry the changing of the technological parameters are very difficult. All parts of the plane have to be produce according to the manuals (standards). In this case for the landing gears it is known that the temperature of the baking heat treatment should not exceed 246°C according to the quoted literature. In the final conclusion of the paper authors have confirmed that the above mentioned temperature is the best for the heat treatment of the landing gear. We could have expected such a result. In my opinion the scientific level of the paper is not high, although I do not have special critical comments concerning description of the research. However, some parts of the paper are close to [26].

In the paper I found some inaccuracies that should be explained and corrected:

1. The references used are sufficient for the paper's issues clarification. However each one (two) of the quoted references should be discussed individually and demonstrate their significance to the work. It is not necessary used four or even five references in one bracket: [1-4], [5-8], [11-17], [18-24].

2. Figs. 5 and 6. To better visualization of the research results Authors should add trend lines.

3. What was an initial size of the rod where from samples were cut off?

4. Authors should add the tensile tests results for STD, SP, SP B200, SP B246 and SP B260 samples. Not only YS, UTS and elongation value. What was an error of the tests?

Author Response

see the attachement

Reviewer 3 Report

In this paper, the authors analyzed the effect of baking heat treatment on the fatigue strength and life of shot peened 4340M material.

From my point of view there are some aspects to improve:

1. Abstract does not contain a clearly method described. Describe briefly the main methods or treatments applied. Also, the abstract should be a single paragraph of about 200 words maximum. Abstract should be revised and improved.

2. The current state of the research field should be reviewed carefully and key publications cited. Please highlight controversial and diverging hypotheses when necessary. The others research should be commented. The Introduction section should be improved.

3. Please cite the reference for the Chemical composition of 4340M steel from Table 1.

4.Please describe in detail the ultrasonic fatigue test not only the basic theory. What kind of amplitude magnifying horn did you used?

5. The micro hardness measurement method should be described on the Methods section not on the Results section. Please move the paragraph ".... for five specimens using MICROMET 3." to the Methods section and clearly explain it.

6. Figure 6, S-N curve obtained from fatigue test should be improved. Please draw the S-N curves, only the points are shown. Also, all the Figures should be on a good resolution.

7.How many samples have been analyzed for all the test? How were manufactured the samples?

8."Fatigue fracture surfaces are shown in Figure. 7”. Please describe in the Method section how the surface was analyzed. What kind of microscope did you use?

9.The quality of the SEM pictures should be improved.

10. The composition analysis procedure of the materials should be mentioned and briefly described in the Method section.

11. Please mention some example of aircrafts that have the landing gear from 4340M material.

12. What are the recommendations of the aircraft manufacturers for the heat treatments of 4340M material. You can answer on it based on the mentioned reference of Boeing Company "Boeing, Standard Overhaul Practices Manual 20-10-03, Revision No.46, 2018".

13. Are the limitations of this study noted? The limitations of this study should be discussed.

14. Please mention the applications of this study.

Author Response

see the attachement

Round 2

Reviewer 2 Report

The authors took into account all the comments of the reviewer and introduced appropriate changes in the text

Reviewer 3 Report

The paper should be improved and it should be well explained for a good reader understanding. Also, some author's answers should be mentioned in the manuscript not only in the cover letter. These will improve the paper. From my point of view there are some aspects to improve:
1. The abstract should contain 200 words maximum based on the journal template. Abstract should be revised.
2.The references should be discussed individually and demonstrate their significance to the work. The Introduction section should be improved based on the above comment.
3. Based on ASTM E8 standard: "4.2 The results of tension tests of specimens machined to standardized dimensions from selected portions of a part or material may not totally represent the strength and ductility properties of the entire end product or its in-service behavior in different environments."
It is unclear how the specimens were manufactured. Was the specimens machined to standardized dimensions from selected portions of a part (aircraft landing gear)?
If the answer is yes, please indicate what kind of aircraft landing gear was used. It is necessary to specified it in the paper for a good reader understanding.
4. All the abbreviations should be explained in the manuscript for a good reader understanding (as example the abbreviations from Table 3).
5. The devices and tools used for this research were not mentioned in the paper.
Please mention in the paper what kind the furnace was used. A picture containing the samples inside the furnace should be inserted in the paper. Also the type of microscope was not mentioned.
6. “Tensile test results were obtained using three specimens according to each condition, respectively” and “Table 4 shows mechanical properties of 4340M steel obtained from tensile test”.
The results from the tensile tests should be statistically analyzed. Thus, the mean values and standard deviations should be mentioned in the Table 4.
7. All the SEM photographs (10 photographs) should be explained for a good reader understanding.
8. The possible applications and the limitations of this study were not mentioned in the paper.
